# AM-PAC 6-Clicks Basic Mobility and Daily Activities Scores Predict 90-Day Modified Rankin Score in Patients with Acute Ischemic Stroke Secondary to Large Vessel Occlusion

**DOI:** 10.3390/jcm13237119

**Published:** 2024-11-25

**Authors:** Andrew Cho, Dhairya A. Lakhani, Aneri B. Balar, Hamza Salim, Manisha Koneru, Argye Hillis, Marlis Gonzalez Fernández, Vaibhav Vagal, Victor Urrutia, Tobias D. Faizy, Jeremy J. Heit, Greg W. Albers, Ishan Mazumdar, Kevin Chen, Sadra Sepehri, Minsoo Kim, Licia Luna, Janet Mei, Vivek S. Yedavalli, Nathan Hyson

**Affiliations:** 1Department of Radiology and Radiological Sciences, Johns Hopkins University, Baltimore, MD 21218, USA; acho27@jh.edu (A.C.); dhairyalakhani@gmail.com (D.A.L.); aneribbalar@gmail.com (A.B.B.); hamza.sleeem@gmail.com (H.S.); vurruti1@jhmi.edu (V.U.); imazumd1@jhmi.edu (I.M.); kchen72@jhmi.edu (K.C.); ssepehr3@jh.edu (S.S.); mkim220@jhmi.edu (M.K.); lluna6@jhmi.edu (L.L.); jmei12@jh.edu (J.M.); nhyson1@jh.edu (N.H.); 2Department of Neuroradiology, Rockefeller Neuroscience Institute, West Virginia University, Morgantown, WV 26505, USA; 3Neuroendovascular Program, Massachusetts General Hospital, Harvard University, Boston, MA 02138, USA; 4Department of Radiology, Cooper Medical School of Rowan University, Camden, NJ 08103, USA; mkoneru2@alumni.jh.edu; 5Department of Neurology, Johns Hopkins University, Baltimore, MD 21218, USA; argye@jhmi.edu; 6Department of Physical Medicine and Rehabilitation, Johns Hopkins School of Medicine, Baltimore, MD 21218, USA; marlis@jhmi.edu; 7Department of Neurology, University of Cincinnati, Cincinnati, OH 45221, USA; vaibhavvagal7@gmail.com; 8Department of Radiology, Neuroendovascular Division, University Medical Center, 37099 Münster, Germany; tobiasfaizy@web.de; 9Department of Neurology, Stanford University, Stanford, CA 94305, USA; jheit@stanford.edu (J.J.H.); albers@stanford.edu (G.W.A.)

**Keywords:** anterior circulation, AM-PAC 6-Clicks, acute ischemic stroke, large vessel occlusion

## Abstract

**Background:** The relative level of functional impairment in stroke patients is a significant determinant of post-acute care. The Activity Measure for Post Acute Care 6-Clicks (AM-PAC) scores for basic mobility and daily activity are rapid standardized assessments whose utilities in predicting long-term stroke outcomes have not yet been studied. **Methods:** We performed a retrospective analysis of acute ischemic stroke patients and their outcomes. We evaluated the distribution of outcomes using the chi-square test. We then compared the proportions of patients with good stroke outcomes for different combinations of favorable scores. We performed two-proportion z-tests to determine the significance, and *p* < 0.05 was considered significant. **Results:** 282 patients met all of the inclusion criteria between 12 January 2017 and 19 March 2023 (*M* age = 66.4, 59.9% female). After dichotomizing the scores as “favorable” vs. “unfavorable”, we found that 128/155 (82.6%) patients with favorable basic mobility had good stroke outcomes vs. 20/127 (15.7%) with unfavorable basic mobility (*p* < 0.0001). Similarly, for favorable daily activity, it was 103/113 (91.2%) vs. 45/169 (26.6%), for both favorable, it was 100/109 (91.7%) vs. 48/173 (27.7%), and for neither favorable, it was 17/123 (13.8%) vs. 131/159 (82.4%), all with *p* < 0.0001. When comparing among groups, both favorable patients differed significantly from those with favorable basic mobility alone (*p* = 0.033) but not those with favorable daily activity alone (*p* = 0.875). Even after adjusting for age, the odds ratios of favorable scores were greater than 20 for any combination (*p* < 0.001). **Conclusions:** Basic mobility and daily activity AM-PAC scores at discharge are independent predictors of anterior circulation acute ischemic stroke outcomes at 90 days.

## 1. Introduction

Stroke is a leading cause of morbidity and mortality worldwide [1]. Acute ischemic stroke secondary to large vessel occlusion accounts for up to 38% of all acute ischemic strokes [2]. Although multiple measures of stroke outcomes exist, the most widely used is the modified Rankin Scale, a seven-point scale, ranging from zero to six, for assessing global disability [3]. The modified Rankin Scale has been validated as a primary long-term outcome measure for stroke intervention at 90 days after hospital discharge [4]. Scores of 0–2 are coded to mean little to no disability with preserved independence. Scores of 3 and above represent stroke outcomes that impact the independence of patients.

Prognosis is a major determinant of post-acute care and discharge planning, a complex decision-making process often requiring multidisciplinary teams communicating with patients and their families [5]. However, methods for predicting stroke outcomes at or around the time of discharge are somewhat limited. The National Institute of Health Stroke Scale (NIHSS) has been found to be an independent predictor of the modified Rankin Scale at 90 days [6]. Although the NIHSS is promising for its speed and decent inter-rater reliability in individual domains, it is reported as a cumulative score with significant variability [7]. The NIHSS has also been criticized for lacking robustness. One study of patients with NIHSS scores of zero showed residual impairment secondary to stroke at 90 days [8]. In addition, the weighting of items included in the NIHSS contributes to bias in evaluations of left- vs. right-sided strokes [9]. For example, language represents up to seven points, while neglect only represents up to two. Furthermore, the NIHSS requires adequate training for its use to be reliable, and it may be limited in its ability to capture functional changes over the course of hospitalization, which may influence prognosis at discharge [10,11]. Other studies have determined that the modified Rankin Scale assessment at discharge and discharge disposition are independent predictors of stroke outcome, with serial measurements offering a potential avenue for further increasing the predictive precision [12,13]. However, because the modified Rankin Scale includes broad categorizations, it does not communicate the specific domains of impairment that the NIHSS captures.

The Activity Measure for Post-Acute Care (AM-PAC) is an assessment of mobility, daily activity, and cognition. The AM-PAC 6-Clicks is a previously validated shortened version of the AM-PAC developed for the basic mobility and daily activity domains [14]. Like the NIHSS, the AM-PAC 6-Clicks is also a rapid assessment tool with good inter-rater reliability [15]. Each questionnaire is composed of six questions rated on a scale of one to four, with total scores ranging from 6, representing severe impairment, to 24, representing no significant impairment. The basic mobility assessment includes items such as ambulation and positional change, whereas the daily activity form includes personal grooming and eating. Both forms measure the range of assistance needed to accomplish such tasks, which can be assessed by clinicians through direct observation or clinical judgment during the hospital course. In contrast to the NIHSS, the AM-PAC 6-Clicks forms are much simpler to administer, which may improve their reliability. Furthermore, because the assessments are reported separately, the AM-PAC 6-Clicks does not necessarily sacrifice precision toward the end of a given hospital course.

Over the last decade, the AM-PAC 6-Clicks has been increasingly studied for its predictive potential. One study determined that both short forms were found to have sensitivities, specificities, positive predictive values, and negative predictive values greater than 70%, with optimized score cutoff thresholds, when used to predict discharge location [16]. Another study found that lower scores in either domain were associated with significantly higher odds of being discharged into skilled nursing facilities or inpatient rehabilitation facilities [17]. Other studies have narrowed the scope of patients to those with cardiovascular disease, those in a cardiac ICU, those receiving total joint arthroplasty, those with traumatic brain injury, and those undergoing adult spinal deformity surgery [18,19,20,21,22,23]. One study found that lower scores were associated with higher rates of short-term readmissions [24]. While these studies, thus far, demonstrate the significant potential of the AM-PAC 6-Clicks, they have focused primarily on discharge location and short-term outcomes. In contrast, the basic mobility and daily activities scores have not yet been studied to predict the long-term outcomes of acute ischemic stroke secondary to large vessel occlusion [25].

In this study, we aim to validate the AM-PAC 6-Clicks basic mobility and daily activity scores as predictors of outcomes of acute ischemic stroke secondary to large vessel occlusions at 90 days, as measured by the modified Rankin Scale. We focus on patients with proximal anterior intracranial strokes, as mechanical thrombectomy has been studied extensively within these vascular territories [25].

## 2. Methods

### 2.1. Study Population

This study included consecutive patients diagnosed with anterior circulation acute ischemic strokes from 12 January 2017 to 19 March 2023 within the Johns Hopkins Hospital system, as well as partnered hospitals within Baltimore, Maryland. Proximal anterior intracranial acute ischemic stroke secondary to large vessel occlusion was defined as an occlusion of the intracranial internal carotid artery or of the M1 or proximal M2 segments of the middle cerebral artery [25]. This study was approved by the Johns Hopkins School of Medicine Institutional Review Board (JHU-IRB00269637).

### 2.2. Inclusion and Exclusion Criteria

Eligible patients were between 18 and 85 years of age of either sex with a diagnosed proximal anterior intracranial acute ischemic stroke secondary to large vessel occlusion. Patients were eligible if they were assessed using both AM-PAC 6-Clicks basic mobility and daily activity scores, as well as followed-up at 90 days using the modified Rankin Scale. Patients who had multifocal infarcts that included vascular territories outside of the anterior circulation were excluded.

### 2.3. Data Collection

Clinical data for each patient were collected by manual chart review. The variables included age, sex, race, basic mobility and daily activity scores at discharge, and modified Rankin Scale score at 90 days.

### 2.4. Study Variables and Outcomes

The primary study variables were the basic mobility and daily activity scores at discharge, and the primary outcome variable was the modified Rankin Scale at 90 days. To evaluate their use for clinical applications, we dichotomized the two domains, defining scores for basic mobility ≥ 17 and daily activity ≥ 19 as favorable assessments based on previously determined cutoffs at which patients were at least 50% impaired [24]. We then compared the relative distributions of modified Rankin Scale scores according to this dichotomy for each domain independently as well as in combination. Finally, we dichotomized the modified Rankin Scale itself, where scores ≤ 2 were defined as good stroke outcomes, and then compared their relative proportions and odds ratios [3].

### 2.5. Statistical Analysis

After dichotomizing the basic mobility and daily activity scores, we used the chi-square test, from the SciPy package in Python 3.10.9, to evaluate the distribution of modified Rankin Scale scores for each domain individually, as well as their combined effect (both favorable vs. exactly one favorable vs. neither favorable). The chi-square test was appropriate despite the low counts because <20% of cells had expected counts < 5. After dichotomizing both variables and outcomes, we performed manual two-proportion *t*-tests for each AM-PAC domain individually (favorable vs. unfavorable), patients with both favorable domains simultaneously (both vs. all others), patients with neither favorable domain (neither vs. all others), and across groups of favorable domains (both favorable vs. favorable basic mobility; both favorable vs. favorable daily activity), comparing the proportion of good stroke outcomes within each group. We then performed a logistic regression with age as a covariate to calculate the odds ratios with 95% confidence intervals.

For demographics, statistical comparisons of age were calculated using two-sample z-tests. Comparisons of sex were calculated using two-proportion t-tests. Comparisons of race were calculated using chi-square tests. The 50 patients who were favorable in exactly one domain were excluded from this analysis because of redundancy.

## 3. Results

### 3.1. Patient Demographics

A total of 787 patients were reviewed, and 161 (20.5%) were excluded because of the lack of an identified large vessel occlusion; 230 (29.2%) were excluded for strokes of or including segments beyond the anterior circulation; 109 (13.9%) lacked either or both AM-PAC 6-Clicks scores or were not followed up at 90 days via the modified Rankin Scale; 282 patients were included in the final study. Of the 282 patients, 113 (40.1%) were male, and 169 (59.9%) were female. The mean age of the patients was 66.4 ± 16.5. Demographics for the total patient sample are summarized in Table 1.

### 3.2. Favorable Basic Mobility and Daily Activity Scores Differentiate Patient Outcomes

We dichotomized the basic mobility and daily activity scores to determine whether a simplified interpretation of these scores could meaningfully differentiate patient outcomes. Based on their respective cutoffs, 155 of 282 (55.0%) had favorable basic mobility scores, 113 of 282 (40.1%) had favorable daily activity scores, 109 of 282 (38.7%) were favorable for both scores, and 123 of 282 (43.6%) were favorable for neither. Fifty (17.7%) were favorable in exactly one domain. The demographics are summarized in Table 2, Table 3, Table 4, Table 5 and Table 6. We found that patients with favorable scores were significantly younger than their counterparts (*p* < 0.001) among all groups. There were no other significant demographic differences among the patient groups with favorable scores (basic mobility vs. daily activity vs. both favorable).

For the favorable basic mobility, daily activity, and double-favorable groups, the distribution is right-skewed with more than half of patients at modified Rankin Scale ≤ 2 (favorable basic mobility: 128/155, 82.6%; favorable daily activity: 103/113, 91.1%; both favorable 100/109, 91.7%; all *p* < 0.001). For their respective unfavorable counterparts, the distribution is left-skewed with most at modified Rankin Scale > 2. For individual domains, the distributions of outcomes appear to intersect at around modified Rankin Scale scores of 3. When looking at the combined effects, the both-favorable and neither-favorable distributions also intersect at around modified Rankin Scale scores of 3, with patients with exactly one favorable domain having a distribution of outcomes in between. For all favorable groups, patients were comparably distributed between modified Rankin Scale scores of 0 and 1 with significant decreases beginning and modified Rankin Scale scores of 2. These results are summarized in Table 7, Table 8 and Table 9.

### 3.3. Dichotomized Basic Mobility and Daily Activity Scores Are Independent Predictors of Good Stroke Outcomes

We then compared the relative proportions of “good” (90 d modified Rankin Scale ≤ 2) and “poor” stroke outcomes as its own dichotomy, and 148 out of the total 282 patients (52.5%) had good stroke outcomes; 128 out of the 155 patients with a favorable basic mobility score (82.6%) had good outcomes at 90 days compared to only 20 out of 127 patients (15.7%) with an unfavorable basic mobility score. In addition, 103 out of 113 with a favorable daily activity score (91.2%) had good outcomes compared to 45 out of 169 (26.6%) with an unfavorable daily activity score; 100 out of 109 patients with both favorable scores (91.7%) had good outcomes compared to 48 out of 173 (27.7%) who were unfavorable in at least one domain; 17 out of 123 (13.8%) patients with neither favorable score had good outcomes compared to 131 of 159 (82.4%) who were favorable in at least one domain. All four comparisons were found to be statistically significant at a *p*-value < 0.001. When comparing the group with both favorable scores vs. either favorable basic mobility or daily activity alone, there was a significant difference only with the favorable basic mobility group but not with the favorable daily activity group. These results are summarized in Table 10 and Table 11 and Figure 1.

### 3.4. AM-PAC Scores Remain Significant Predictors After Adjusting for Age

Because age was found to be significantly different within respective group comparisons, we used linear regression with age as a covariate. We converted the log-likelihood statistics to odds ratios for having “good” stroke outcomes for each group. The analysis revealed that having a favorable basic mobility score at discharge was associated with 22.96 times higher odds of having a “good” outcome, favorable daily activity was associated with 24.75 times higher odds, both with 25.02, and neither with 0.04 (all *p* < 0.001). Age was a significant contributor in all cases. These results are summarized in Table 12.

## 4. Discussion

This study demonstrated that both the basic mobility and daily activity domains are independent predictors of outcomes at 90 days for patients with proximal anterior intracranial acute ischemic stroke secondary to large vessel occlusion. To our knowledge, our study is the first to attempt to validate the AM-PAC 6-Clicks shortened forms for basic mobility and daily activity in the acute ischemic stroke population. After simplifying to “favorable” vs. “unfavorable” scores, we found significant differences in the distributions of patient outcomes even after adjusting for age. These results indicate that the AM-PAC 6-Clicks may provide robust predictive value for long-term outcomes. However, clinical applications may be more meaningful when used to simply predict “good” vs. “poor” stroke outcomes. Given this, our results ultimately imply that patients who suffer less than 50% functional impairment at the time of discharge in either mobility or in accomplishing activities in daily living are more likely to retain significant independence 90 days following an ischemic stroke. These results also have interesting implications for the use of the basic mobility and daily activity scores as predictors. As expected, the dichotomized both-favorable patients had the highest proportion of good stroke outcomes, and the neither-favorable had the lowest. However, the both-favorable group had similar outcomes to patients with only one favorable daily activity score (91.7% vs. 91.2%), while the neither-favorable group was similar in distribution to those in the basic mobility score group (13.8% vs. 15.7% unfavorable basic mobility). These similarities suggest that the basic mobility and daily activity domains may have divergent predictive utility. One interpretation is that a favorable daily activity score alone is sufficient for predicting good stroke outcomes, whereas an unfavorable basic mobility score alone is sufficient for predicting poor stroke outcomes. Patients with a favorable daily activity score and unfavorable basic mobility score or vice versa may have more equivocal outcomes. A previous study similarly found differences in the predictive power of the two domains [16]. At the determined optimal cutoffs for sensitivity, specificity, positive predictive value, and negative predictive value in predicting discharge location in a broad range of patients, the basic mobility score was found to be more sensitive and less specific than the daily activity score. The basic mobility score was also found to have the higher negative predictive value, whereas the daily activity score had the higher positive predictive value. It is therefore possible that these differences extend to predicting outcomes as well.

Alternatively, these findings could reflect a redundancy between the domains. Out of 282 patients, only 50 (17.7%) had either favorable basic mobility or daily activity scores alone. Of these 50, 46 (92%) had only a favorable basic mobility score compared to 4 (8%) with just a favorable daily activity score. This implies that it is relatively rare for a patient to retain independence in daily activities with moderate-to-severe impairments of mobility, while the converse is not true. This further supports the possibility that a favorable daily activity score alone may be sufficient to predict good stroke outcomes. The implication of this conclusion would be for discharge planning to prioritize improving patient ability to perform daily activities.

These results bridge previous work in predicting favorable AM-PAC for patients with acute ischemic stroke secondary to large vessel occlusion. It was previously determined that lower initial stroke severity, smaller core infarct volumes, and excellent recanalization via mechanical thrombectomy were predictive of favorable basic mobility and daily activity scores at discharge [26]. This study, therefore, validates these measures during patients’ hospital courses as predictors of eventual long-term outcomes. Future studies could determine the relationship between these factors and the observed change in AM-PAC scores over the course of hospital treatment.

Furthermore, among the variables we addressed in this study, age was a significant factor among groups across all comparisons. Patients with favorable scores were, on average, 10 years younger than their counterparts. This is consistent with younger age having been previously determined to be predictive of favorable AM-PAC scores, as well as older age being associated with worse outcomes overall [26,27]. After adjusting for age, we found that AM-PAC scores were still significant, independent predictors of outcomes.

The degree to which AM-PAC scores were significant predictors was surprising. Even after adjusting for age, the odds ratios of having favorable scores with regard to “good” stroke outcomes was over 20. This further underscores the importance of physical and occupational therapy for patients following a stroke. A previous study found that the majority of patients do not see a therapist after being discharged home [28]. In conjunction with our study, this highlights not only a large unmet patient need but also the severity of its consequences. This issue is further complicated by differences in hospital settings regarding the type of therapy (physical vs. occupational) as well as timeliness of care [29,30]. The AM-PAC 6-Clicks may, therefore, be useful for setting definitive goals for clinicians who are planning discharges for stroke patients to pursue rehabilitation options more proactively.

There are several limitations to this study. First, our study is inherently limited by a retrospective design. Second, we accounted for only a few demographic variables. For example, we did not evaluate comorbidities that may differ between groups, such as birth defects that could alter patient independence at baseline or known contributors to stroke outcomes such as hypertension and hypercholesterolemia [1]. These factors could account for some of the observed associations of stroke outcomes with AM-PAC scores. Future study of the relationship between these variables and their ability to predict AM-PAC scores could further validate the use of the AM-PAC 6-Clicks as predictors of long-term outcomes. Third, our outcome variable for modified Rankin Scale assessments at 90 days ignores many factors that may affect management both in the hospital and after discharge. This includes prehospital delay, subsequent follow-up visits, acute events, or readmissions following discharge from the hospital. Prehospital delay is of particular interest because it includes patient factors that may affect whether a patient is a candidate for reperfusion therapy, which significantly affects prognosis. These include the patient’s location at the time of onset, prior knowledge of stroke as a medical emergency, and language barriers [31]. The modified Rankin Scale also does not consider the potential effects of discharge location or use of rehabilitation services. Our study was intended to evaluate the cumulative outcome of stroke patients, and these factors may be used to stratify such patients in future studies for improving the predictive potential of the AM-PAC 6-Clicks in stroke patients. Finally, although AM-PAC scores have good inter-rater reliability, we did not consider any factors regarding administration, such as clinician and therapist training.

Overall, we conclude that the AM-PAC 6-Clicks basic mobility and daily activity scores are promising adjuncts to existing methods of predicting long-term stroke outcomes.

## Figures and Tables

**Figure 1 jcm-13-07119-f001:**
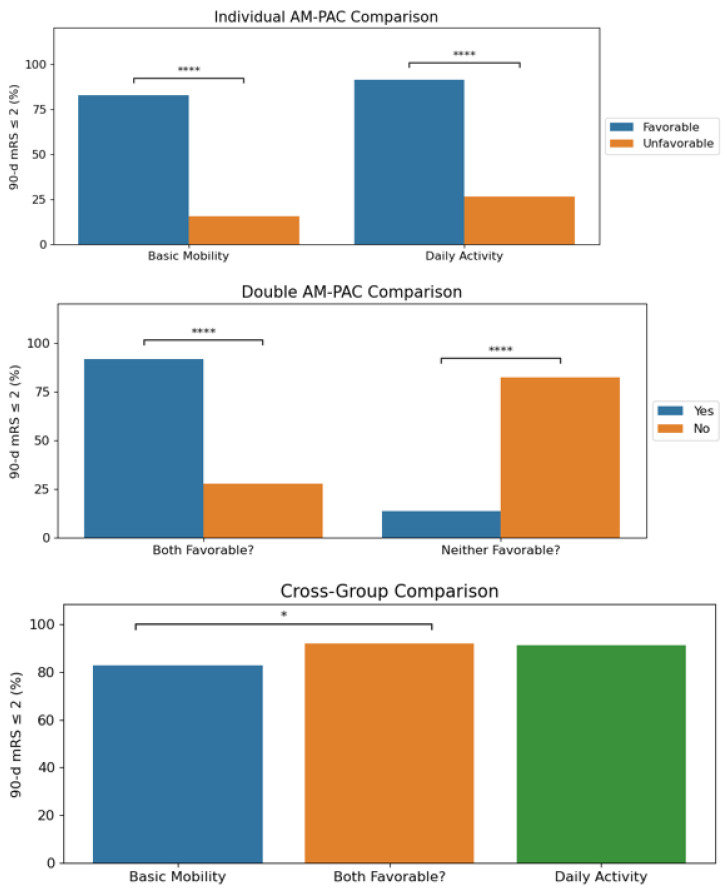
**Basic mobility and daily activity are independent predictors of stroke outcomes.** Significant differences were found within each comparison group regarding good stroke outcomes (reported as a proportion). Within the cross-group analysis, the both-favorable group differed significantly from the favorable basic mobility group but not from the favorable daily activity group. * *p* < 0.05 and **** *p* < 0.0001.

**Table 1 jcm-13-07119-t001:** Total patient demographics.

All Patients (*n* = 282)	
**Age**	66.4 ± 16.5
**Sex**	113 Male (40.1%)
	169 Female (59.9%)
**Race**	120 Black (42.6%)
	141 White (50.0%)
	21 Other (7.4%)

**Table 2 jcm-13-07119-t002:** Patient demographics by basic mobility score. ***: statistically significant, *p* < 0.001.

Basic Mobility Score			
Variable	Favorable (*n* = 155)	Unfavorable (*n* = 127)	*p*-Value
Age	62.7 ± 15.9	71.0 ± 16.1	<0.001 ***
Sex	59 Male (38.1%) 96 Female 61.9%)	54 Male (42.5) 73 Female (57.5%)	0.447
Race	63 Black (40.6%) 76 White (49.0%) 16 Other (10.3%)	57 Black (44.9%) 65 White (51.2%) 5 Other (3.9%)	0.124

**Table 3 jcm-13-07119-t003:** Patient demographics by daily activity score. ***: statistically significant, *p* < 0.001.

Daily Activity Score			
Variable	Favorable (*n* = 113)	Unfavorable (*n* = 169)	*p*-Value
Age	60.8 ± 16.2	70.2 ± 15.7	<0.001 ***
Sex	38 Male (33.6%) 75 Female (66.4%)	75 Male (44.4%) 94 Female (55.6%)	0.070
Race	46 Black (40.7%) 58 White (51.3%) 9 Other (9.0%)	74 Black (43.8%) 83 White (49.1%) 12 Other (7.1%)	0.867

**Table 4 jcm-13-07119-t004:** Patient demographics by both favorable scores. ***: statistically significant, *p* < 0.001.

Both Favorable			
Variable	Both (*n* = 109)	All Others (*n* = 173)	*p*-Value
Age	60.4 ± 16.2	70.2 ± 15.6	<0.001 ***
Sex	37 Male (33.9%) 72 Female (66.1%)	76 Male (43.9%) 97 Female (56.1%)	0.095
Race	45 Black (41.3%) 55 White (50.5%) 9 Other (8.3%)	75 Black (43.4%) 86 White (49.7%) 12 (6.9%)	0.891

**Table 5 jcm-13-07119-t005:** Patient demographics by neither favorable score. ***: statistically significant, *p* < 0.001.

Neither Favorable			
Variable	Neither (*n* = 123)	All Others (*n* = 159)	*p*-Value
Age	71.0 ± 16.3	62.9 ± 15.9	<0.001 ***
Sex	53 Male (43.1%) 70 Female (56.9%)	60 Male (37.7%) 99 Female (62.3%)	0.363
Race	56 Black (45.5%) 62 White (50.4%) 5 Other (4.1%)	64 Black (40.3%) 79 White (49.7%) 16 Other (10.1%)	0.149

**Table 6 jcm-13-07119-t006:** Patient demographics across groups of favorable scores.

Cross-Group (Pairwise Comparisons)					
Variable	Favorable Basic Mobility (*n* = 155)	Favorable Daily Activity (*n* = 113)	Both Favorable (*n* = 109)	*p*-Value	
Age	62.7 ± 15.9	60.8 ± 16.2	60.4 ± 16.2	Both vs. basic mobility:	0.252
				Both vs. daily activity:	0.854
Sex	59 Male (38.1%) 96 Female (61.9%)	38 Male (33.6%) 75 Female (66.4%)	37 Male (33.9%) 72 Female (66.1%)	Both vs. basic mobility:	0.490
				Both vs. daily activity:	0.960
Race	63 Black (40.6%) 76 White (49.0%) 16 Other (10.3%)	46 Black (40.7%) 58 White (51.3%) 9 Other (9.0%)	45 Black (41.3%) 55 White (50.5%) 9 Other (8.3%)	Both vs. basic mobility:	0.852
				Both vs. daily activity:	0.991

**Table 7 jcm-13-07119-t007:** Modified Rankin Scale distribution by dichotomized basic mobility score. ***: statistically significant, *p* < 0.001.

*p*-Value		Modified Rankin Scale						
*p* < 0.001 ***		0	1	2	3	4	5	6
**Basic Mobility**	**Favorable** **(*n* = 155)**	46 (29.7%)	50 (32.3%)	32 (20.6%)	15 (9.7%)	7 (4.5%)	1 (0.6%)	4 (2.6%)
	**Unfavorable (*n* = 127)**	1 (0.8%)	7 (5.5%)	12 (9.4%)	12 (9.4%)	38 (29.9%	11 (8.7%)	46 (36.2%)

**Table 8 jcm-13-07119-t008:** Modified Rankin Scale distribution by dichotomized daily activity score. ***: statistically significant, *p* < 0.001.

*p*-Value		Modified Rankin Scale						
*p* < 0.001 ***		0	1	2	3	4	5	6
**Daily Activity**	**Favorable** **(*n* = 113)**	40 (35.4%)	40 (35.4%)	23 (20.4%)	4 (3.5%)	3 (2.7%)	0 (0%)	3 (2.7%)
	**Unfavorable (*n* = 169)**	7 (4.1%)	17 (10.1%)	21 (12.4%)	23 (13.6%)	42 (24.9%)	12 (7.1%)	47 (27.8%)

**Table 9 jcm-13-07119-t009:** Modified Rankin Scale distribution by combined dichotomized scores. ***: statistically significant, *p* < 0.001.

*p*-Value	Modified Rankin Scale						
*p* < 0.001 ***	0	1	2	3	4	5	6
**Both Favorable** **(*n* = 109)**	40 (36.7%)	39 (35.8%)	21 (19.3%)	4 (3.7%)	3 (2.8%)	0 (0%)	2 (1.8%)
**Exactly One Favorable** **(*n* = 50)**	6 (12%)	12 (24%)	13 (26%)	11 (22%)	4 (8%)	1 (2%)	3 (6%)
**Neither Favorable** **(*n* = 123)**	1 (0.8%)	6 (4.9%)	10 (8.1%)	12 (9.8%)	38 (30.9%)	11 (8.9%)	45 (36.6%)

**Table 10 jcm-13-07119-t010:** Individual comparisons by proportions of good stroke outcomes. ****: statistically significant, *p* < 0.0001.

AM-PAC Group	90-Day Modified Rankin Scale ≤ 2 (%)	*p*-Values
Favorable basic mobility	128/155 (82.6%)	<0.001 ****
Unfavorable basic mobility	20/127 (15.7%)	
Favorable daily activity	103/113 (91.2%)	<0.001 ****
Unfavorable daily activity	45/169 (26.6%)	
Both favorable	100/109 (91.7%)	<0.001 ****
At least one unfavorable	48/173 (27.7%)	
Neither favorable	17/123 (13.8%)	<0.001 ****
At least one favorable	131/159 (82.4%)	

**Table 11 jcm-13-07119-t011:** Cross-group comparisons by proportions of good stroke outcomes. *****: statistically significant, *p* < 0.05.

AM-PAC Group	90-Day Modified Rankin Scale ≤ 2	*p*-Values	
Favorable basic mobility	128/155 (82.6%)	0.032 *	
Both favorable	100/109 (91.7%)		0.873
Favorable daily activity	103/113 (91.2%)		

**Table 12 jcm-13-07119-t012:** Odds ratios of good outcomes with age as a covariate. *: statistically significant, *p* < 0.05; **: statistically significant, *p* < 0.01; ***: statistically significant, *p* < 0.001.

AM-PAC Group	Odds Ratios (95% CI)	*p*-Value
Favorable basic mobility	22.96 (12.07–43.68)	<0.001 ***
Age	0.97 (0.95–0.99)	0.005 **
Favorable daily activity	24.75 (11.81–51.88)	<0.001 ***
Age	0.98 (0.96–0.99)	0.012 *
Both favorable	25.02 (11.63–53.81)	<0.001 ***
Age	0.98 (0.96–1.00)	0.016 *
Neither favorable	0.04 (0.02–0.07)	<0.001 ***
Age	0.97 (0.96–0.99)	0.004 **

## Data Availability

The datasets used and/or analyzed during the current study are available from the corresponding author upon reasonable request.

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
