# Peer review of "AM-PAC 6-Clicks Basic Mobility and Daily Activities Scores Predict 90-Day Modified Rankin Score in Patients with Acute Ischemic Stroke Secondary to Large Vessel Occlusion"

_jcm, 2024, doi:10.3390/jcm13237119_

Round 1
Reviewer 1 Report
Comments and Suggestions for Authors
The abstract is too long, ideally it should be kept to 250 words
The introduction provides sufficient information, however the first part that refers to the modified rankin scale is too long and extensive. It is not necessary for the authors to indicate what each score obtained corresponds to. However, it would be necessary to correlate the symptoms and sequelae produced by the stroke and its study.
In the methods it is necessary to put the registration number of the relevant ethics committee.The rest of the sections of the methods are explained correctly.
The results are described adequately, showing information about each item. It would be interesting if the authors included the relationship that there may be between training and chronic birth defects in this correlation.
The discussion must be worked on. The authors only provide two different bibliographical references to the introduction, surely there are many more that they can provide to discuss their study. In addition, they should also include a paragraph on clinical implications.
The bibliographic references are not correctly inserted, correct the numbers that are inserted twice.
Author Response
The abstract is too long, ideally it should be kept to 250 words
We agree that the abstract could be condensed. We have shortened it to be less than 250 words.
The introduction provides sufficient information, however the first part that refers to the modified rankin scale is too long and extensive. It is not necessary for the authors to indicate what each score obtained corresponds to. However, it would be necessary to correlate the symptoms and sequelae produced by the stroke and its study.
We agree that the explanation for the modified Rankin Scale was irrelevant in its detail. We have replaced that section with just a broad summary of its meaning [lines 94 – 105].
In the methods it is necessary to put the registration number of the relevant ethics committee.The rest of the sections of the methods are explained correctly.
We have now added the appropriate IRB registration number [lines 164-165].
The results are described adequately, showing information about each item. It would be interesting if the authors included the relationship that there may be between training and chronic birth defects in this correlation.
We appreciate the suggestion and have included the potential effects of training and chronic birth defects in our discussion about directions for future study [lines 415 – 417, 431 – 432].
The discussion must be worked on. The authors only provide two different bibliographical references to the introduction, surely there are many more that they can provide to discuss their study. In addition, they should also include a paragraph on clinical implications.
We have added more references and bolstered areas of the discussion with larger emphasis on clinical implications [lines 400 – 432].
The bibliographic references are not correctly inserted, correct the numbers that are inserted twice.
We have now adjusted the references based on the extended MDPI reference guide. Please let us know if there are any other issues.
Reviewer 2 Report
Comments and Suggestions for Authors
A retrospective analysis of patients with a) acute ischemic stroke secondary to large vessel occlusion in the anterior circulation (intracranial ICA, proximal M1 and M2), b) AM-PAC scores for basic mobility and daily activity assessed at discharge, and c) stroke outcomes measured at 90 days using the modified Rankin Scale. We visualized the relationship between AM-PAC scores and the modified Rankin Scale using a heat map. Spearman's rank correlation test is used to determine statistical significance. Favorable basic mobility and daily activity scores are defined as ≥ 17 and ≥ 19, respectively.
The study is very interesting.
Aspects to consider.
Enter this citation: PubMed ID 31627368
Figures 1 and 2, I don't know if they provide information, consider them.
I don't know if tables 7, 8, and 9 are relevant to the percentage in each score. The possibility of expressing them differently should be studied.
Age appears as a very important factor in people with the worst score. Consider it as a covariate.
Essential, table 10 should be an ancova with the dependent variable being the post-test differential score, minus the pre-test, and the covariate being the pre-test, and the independent variable being the group.
Author Response
A retrospective analysis of patients with a) acute ischemic stroke secondary to large vessel occlusion in the anterior circulation (intracranial ICA, proximal M1 and M2), b) AM-PAC scores for basic mobility and daily activity assessed at discharge, and c) stroke outcomes measured at 90 days using the modified Rankin Scale. We visualized the relationship between AM-PAC scores and the modified Rankin Scale using a heat map. Spearman's rank correlation test is used to determine statistical significance. Favorable basic mobility and daily activity scores are defined as ≥ 17 and ≥ 19, respectively.
The study is very interesting.
Aspects to consider.
Enter this citation: PubMed ID 31627368
We appreciate the citation and have included it in our discussion of limitations [lines 424 – 427].
Figures 1 and 2, I don't know if they provide information, consider them.
We agree that the previous section 3.1 as well as Figures 1 and 2 were redundant for the overall findings of the paper. We have removed them.
I don't know if tables 7, 8, and 9 are relevant to the percentage in each score. The possibility of expressing them differently should be studied.
We believe that the relative proportions of each outcome are useful for summarizing the differences in distribution of outcomes between each group.
Age appears as a very important factor in people with the worst score. Consider it as a covariate.
Essential, table 10 should be an ancova with the dependent variable being the post-test differential score, minus the pre-test, and the covariate being the pre-test, and the independent variable being the group.
We agree with the suggestion to consider age as a covariate. We previously discussed the issue of age briefly in the discussion section, but we have now added a new section in the results (3.4) for more statistical analysis. We believe that Table 10 is still useful for demonstrating the raw differences between groups, but we now have Table 12 with the results of logistic regression with age as a covariate [lines 334 – 344].
Round 2
Reviewer 1 Report
Comments and Suggestions for Authors
Delete the word title from the title of the paper
The authors have made the improvement modifications that have been requested, providing the document with high quality
Reviewer 2 Report
Comments and Suggestions for Authors
The authors have made the suggested changes, except for the ancova, I leave it to the editor's decision to evaluate this aspect for publication.